# Using Hyperspatial LiDAR and Multispectral Imaging to Identify Coastal Wetlands Using Gradient Boosting Methods

**Shitij Govil** [1,*], **Aidan Joshua Lee** [1], **Aiden Connor MacQueen** [1], **Narcisa Gabriela Pricope** [2], **Asami Minei** [2] **and Cuixian Chen** [3]

1. Marvin Ridge High School, 2825 Crane Rd, Waxhaw, NC 28173, USA
2. Department of Earth and Ocean Sciences, University of North Carolina Wilmington, 601 S College Rd., Wilmington, NC 29403, USA
3. Department of Mathematics and Statistics, University of North Carolina Wilmington, 601 S College Rd., Wilmington, NC 29403, USA
* Correspondence: shitijgovil@gmail.com

**Abstract:** Wetlands play a vital role in our ecosystems, preserving water quality, controlling flooding, and supplying aquifers. Wetlands are rapidly degrading due to threats by human encroachment and rising sea levels. Effective and timely mapping of wetland ecosystems is vital to their preservation. Unoccupied Aircraft Systems (UAS) have demonstrated the capability to access and record data from difficult-to-reach wetlands at a rapid pace, increasing the viability of wetland identification and classification through machine learning (ML) methods. This study proposes a UAS-based gradient boosting approach to wetland classification in coastal regions using hyperspatial LiDAR and multispectral (MS) data, implemented on a series of wetland sites in the Atlantic Coastal Plain region of North Carolina, USA. Our results demonstrated that Xtreme Gradient Boosting performed the best on a cross-site dataset with an accuracy of 83.20% and an Area Under Curve (AUC) score of 0.8994. The study also found that Digital Terrain Model-based variables had the greatest feature importance on a cross-site dataset. This study's novelty lies in utilizing cross-site validation using Gradient Boosting methods with limited amounts of UAS data while explicitly considering topographical features and vegetation characteristics derived from multi-source UAS collections for both wetland and non-wetland classes. Future work is encouraged with a larger dataset or with semi-supervised learning techniques to improve the accuracy of the model.

**Keywords:** wetland classification; gradient boosting; UAS; machine learning

## 1. Introduction

In the 1600s, there were an estimated 392 million acres of wetlands recorded in the United States, 221 million of which were in the contiguous states alone. Today, over half of the original wetland cover in the lower 48 states has been drained and converted for other land uses [1]. At the same time, nearly 70% of worldwide wetlands have been lost since the 1900s [2]. With continued rapid urbanization in coastal regions across the world, wetlands are at heightened risk of degradation [3]. Wetlands play an essential role in the hydrological cycle, often reducing floods, recharging groundwater aquifer sources, and augmenting low flows during dry periods [4]. They also play a vital role in a variety of other ecological services including preserving water quality, providing habitat for fish and wildlife, preventing erosion, and providing aesthetically pleasing spaces for recreational activity [5].

Although wetlands are degrading at a rapid pace, the Environmental Protection Agency (EPA) has recognized both the importance and vulnerability of these ecosystems and moved to protect them with the Clean Water Act, signed in 1972. This act prohibits the discharge of pollutant materials into the "waters" which includes wetlands, of the United States [6]. Although regulations that afford protection to wetlands have provided some

degree of protection against degradation and replacement, wetlands that have not yet been classified cannot be recognized by the Clean Water Act and remain vulnerable to human encroachment. The most reliable system for classifying and protecting wetlands in the United States is the National Wetlands Inventory (NWI) system, created by the U.S. Fish and Wildlife Service in response to an order from the Emergency Wetlands Resources Act of 1986 [7]. Yet, in many places in the country, the maps produced by the NWI are severely outdated and lacking in precision and accuracy [8].

Oftentimes, wetlands are classified in the field using wetland classification systems such as field guides, delineation manuals, and other tools [9–11]. Additionally, many wetlands are classified via evaluation of their hydrology, vegetation, and hydric soils [12]. This approach, however, is often impractical on larger scales. Field-based methods are frequently infeasible due to the time and cost associated with the process [13]. Additionally, many wetlands are located in rural, topographically isolated areas making field visitation challenging and impractical [14]. Additionally, it should be noted that given the temporal variability of wetlands, visiting the sites frequently is necessary to ensure the accuracy of wetland inventories.

To tackle these challenges in field-based classification, researchers are finding new ways of accurately collecting data and classifying wetlands using remote sensing technology. For decades, various machine learning methods (such as K Nearest Neighbors, Decision Trees, and Support Vector Machines) have been used to classify wetlands using remote sensing data [15–18]. In more recent years, Convolutional Neural Networks (CNNs), as well as other Artificial Neural Networks (ANNs), have been used to analyze images to classify wetland cover [19–21]. However, CNNs are plagued by high computational costs and a tendency to overfit with limited amounts of training data [22]. Ensemble tree-based methods have recently gained popularity for land cover classification, especially due to their high accuracy [23]. Additionally, they reduce the risk of choosing a poor classifier by averaging the votes of individual classifiers. Random forests have been successful with their performance due to their high accuracy [24–26]. Meanwhile, methods like Gradient Boosting have seen great usage and success in the field of remote sensing with both satellite and drone-based imagery [27–30]. Gradient Boosting works by sequentially building weak classification algorithms based on the residuals of previous algorithms, improving based on observations that are poorly predicted by predecessors [31,32].

These classifications, which have been conducted primarily on satellite-based remote sensing data collected by Landsat or Sentinel satellites, have their limitations [33]. Past literature has used Landsat or Sentinel-2b for standardized global topographic data using Multi-spectral Instruments (MSI) [34]. Satellite-based datasets only perform at a 10 m resolution, limiting the amount of data available [33]. Current research seeks to use pixel-based datasets with finer resolutions to increase the sample size and improve the performance of wetland classification [35].

Unoccupied Aerial Systems (UAS) have risen to prominence as effective tools for remote sensing due to the high costs and barriers to using satellite-based imagery [36]. They also serve as alternatives to occupied aircraft systems, which are limited by their cost and complexity [37]. The removal of these barriers allows for more on-demand flight control for independent researchers [38]. Cloud cover is also a major detriment to using satellite-based imagery, as clouds can often obscure topographical features from satellites, but UAS can fly under the clouds and collect topographical data [39].

UAS are being combined with various sensors such as RGB cameras for visible spectrum data, Hyperspatial Light Detection and Ranging Sensors (LiDAR), Multispectral Sensors (MS), and Synthetic Aperture Radar (SAR) [32,40,41]. The advantage that LiDAR data offers is a high resolution that allows more data to be drawn from a smaller site, improving performance on smaller sites [42]. Data collected from the drone-based sensors are often used in ensemble-based Machine Learning (ML) algorithms such as Gradient Boosting and its variants such as Extreme Gradient Boosting and Light Gradient Boosting [43–46].

A majority of past literature using Gradient Boosting to classify wetlands utilizes satellite-based imagery [32,40,47]. These studies, however, are restricted to one site and only train the model on wetland land-cover, not considering non-wetland land-cover. Additionally, these studies use large amounts of training data (over 10,000 samples). Some studies have used a combination of both satellite and UAS data to classify wetlands using Gradient Boosting while others consider only one study site in the development of a model [48].

This study aims to investigate whether Gradient Boosting used on LiDAR and Multispectral data is an effective way of identifying wetlands through UAS-based remote sensing. This study's novelty lies in using cross-site validation with UAS-collected data while explicitly considering the topographical and vegetation-based variables of non-wetlands for training the model. Two objectives are proposed:

(1) Explore the effectiveness of different Gradient Boosting models used on LiDAR and Multispectral data for identifying wetland and non-wetland classes across multiple sites.
(2) Determine the best variables to classify wetlands using Gradient Boosting.

## 2. Materials and Methods

### 2.1. Study Areas

Data at seven sites (the Masonboro site was sampled twice based on tide conditions) shown in Table 1 and Figure 1 were collected in a Coastal Plain region in Southeastern North Carolina, United States using UAS and ground surveys. These wetlands sites were chosen to include various types of wetlands, such as Palustrine Forested (PFO), Palustrine Scrub-Shrub (PSS), Palustrine Emergent (PEM), Unknown Perennial Riverine Unconsolidated Bottom (R5UB), Intertidal Estuarine Emergent (E2EM), Marine Intertidal Unconsolidated Shore (M2US), and Subtidal Estuarine Unconsolidated Bottom (E1UB). Both UAV LiDAR and multispectral data, as well as ground reference habitat points were collected during same-day surveys at all sites between May 2020 and January 2021.

**Table 1.** List and Characteristics of Study Areas.

| Site Name | Coordinates | Wetland Types | Date Surveyed | Wetland Habitat Points | Non-Wetland Habitat Points | Area of Interest (Acres) | Spatial Resolution (cm) |
|---|---|---|---|---|---|---|---|
| St. James | 78°8′22″W 33°57′7″N | PFO, PSS | 12 May 2020 | 80 | 28 | 178.00 | 12.69 |
| Topsail | 77°41′6″W 34°23′51″N | PFO, PSS | 2 June 2020 | 47 | 43 | 113.27 | 11.99 |
| Castle Bay | 77°42′18″W 34°24′42″N | PFO, PSS, PEM, R5UB | 29 June 2020 | 91 | 70 | 128.45 | 12.00 |
| River Road | 77°55′11″W 34°5′12″N | PFO, PEM, E1UB | 2 October 2020 | 95 | 40 | 54.34 | 12.50 |
| Surf City | 77°33′15″W 34°26′24″N | PFO, PSS, E2EM, E1UB | 6 November 2020 | 96 | 61 | 78.28 | 12.61 |
| Masonboro High Tide | 77°49′39″W 34°10′15″N | E1UB, E2US, M2US, E2EM | 7 December 2020 | 97 | 56 | 109.98 | 12.44 |
| Masonboro Low Tide | 77°49′39″W 34°10′15″N | E1UB, E2US, M2US, E2EM | 11 December 2020 | 96 | 56 | 109.98 | 12.44 |
| Maysville | 77°14′17″W 34°54′1″N | PFO, R5UB | 22 January 2021 | 27 | 53 | 43.80 | 12.69 |

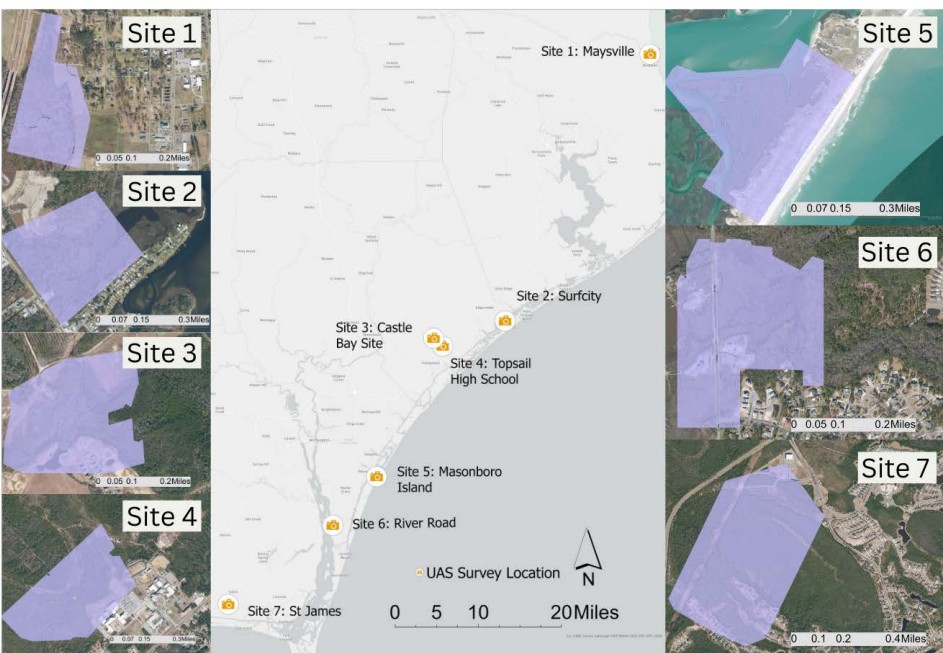

**Figure 1.** Study area in southeastern North Carolina, United States and the associated survey locations for each site.

### 2.2. Data Collection and Preprocessing

The field collected data used in this research were acquired using a DJI Matrice 600 Pro drone outfitted with a Quanergy M8 LiDAR sensor that collected hyperspatial LiDAR data and a fixed-wing SenseFly eBee Plus drone with a Parrot Sequoia sensor that collected multispectral data. The Parrot Sequoia sensor can detect green, red, red-edge, and near-infrared wavelengths of light, which are typically used to classify vegetation and ground soil [49]. The difference between the data collected using UAS LiDAR and Sentinel-2b and RapidEye imagery is that satellite cover may be blocked by topographic weather conditions, and it may be difficult to procure the most recent data. It is also difficult to add active sensors (radar imagery) to Sentinel-2b, adding restrictions to the elevation variables that can be measured [50]. The difference between these two datasets is that hyperspatial LiDAR is a newer technology and is capable of higher quality resolutions of less than 1 cubic meter [41].

Ground reference data for each wetland and non-wetland land cover type (hereby called habitat points) were collected on the same day as the aerial surveys using a Trimble R10 GNSS RTK System. Where habitat points that had been randomly generated prior to fieldwork were not accessible, they were supplemented using the 2020 National Agriculture Imagery Program (NAIP) as described in detail in Pricope et al., 2022 [41]. Habitat points were used to train the model and were randomly selected to produce roughly 15 samples of each wetland type and 50 samples of non-wetlands for distinguishing among the land cover types present in the surveys.

For each of the surveyed sites, two flights were conducted to minimize within site variability and tidal and environmental conditions changes as much as possible. The UAS LiDAR surveys were referenced using static and aircraft GNSS data, both collected using a Trimble RTK device as described in detail in Pricope et al., 2022. Data from the Internal Navigation System (INS) and the Global Navigation Satellite System (GNSS) collected by the GPS receiver were then downloaded and converted to a text file. After the file was converted, spatial and distance filtering settings were applied to remove noise, and then converted into point cloud data. Points that did not have Ground Control Points (GCPs) could not be georeferenced and were excluded from the dataset. The constrained point clouds, however, had corresponding GCPs and could be georeferenced. The data was

further processed to remove any noise, such as birds or leaves under the drone. Known ground surveyed points in the area were then compared to all nearby LiDAR scanned points in terms of elevation. Lastly, the elevation of the points was changed to the average of the known RTK ground points and the LiDAR points to compute the final elevation of each point. For a detailed description of the LiDAR preprocessing workflow, please see Pricope et al., 2022 [41].

Sequoia Multispectral Data was preprocessed in two parts, the first being Post-Processed Kinematic correction (PPK), and the second being image processing [50]. The PPK correction method used location data collected from the onboard eBee system to track the position of the drone at the time each picture was taken and accurately position each of the images at their location. Through image processing, the location of each of the images was considered and compared to target radiometric calibration images taken at the same location and during the same environmental conditions. The same GCPs used in the LiDAR processing workflow were used to georeference the multispectral imagery with an accuracy of 10–27 cm [50]. The result from the preprocessing of the multispectral images created a TIFF image file that included 4 raster layers for green, red, red edge, and near-infrared reflectance values.

The processed image is a pixel grid, where each pixel has a set of variables mapped to it. Each dataset consists of topographical features shown in Table 2, as well as a variable representing whether the area is a wetland, which is collected from the habitat points sampling. The multispectral sensor is used to map the mass and density of vegetation in a sampled area.

**Table 2.** Summary showing the predictors and response variables derived from the UAS-collected LiDAR and multispectral data.

| Raster Layer | Variable | Data Input | Definition |
|---|---|---|---|
| 1 | DSM | Digital Surface Model | Max height elevation in meters (including vegetation and artificial objects) [51] |
| 2 | DTM | Digital Elevation Model | Ground elevation in meters (vegetation and artificial objects removed) [52] |
| 3 | sDTM | Smoothed DTM | Ground elevation in meters, where microtopographic noise is removed [53] |
| 4 | hDTM | Hydro-condition DTM | Hydro-conditioning resolves topographic depressions before modeling flow paths [54] |
| 5 | Aspect | Aspect | Compass direction of the steepest downhill gradient [55] |
| 6 | Slope | Slope | The rate of change of elevation per DTM cell [56] |
| 7 | Curvature | Curvature | Combined curvature value from PlanCurv and ProfileCurv [57] |
| 8 | PlanCurv | Plan Curvature | The horizontal curvature of the slope [57] |
| 9 | ProfileCurv | Profile Curvature | The vertical curvature of the slope [57] |
| 10 | NDVI | Normalized Difference Vegetation Index | Uses the contrast of vegetation between near-infrared and red light to calculate the relative biomass in an area [58] |
| 11 | NDWI | Normalized Difference Water Index | An index that is used to measure the water content in vegetation at the canopy level using the green (550 nm $\pm$ 40 nm) and near-infrared band (790 nm $\pm$ 40 nm) reflectance values based on the McFeeters NDWI Index [59] |
| 12 | NDRE | Normalized Difference Red Edge Index | Measures the relative chlorophyll in plants due to reflecting light using the red edge (735 nm $\pm$ 10 nm) and near-infrared (790 nm $\pm$ 40 nm) band reflectance values [60] |
| 13 | CHM | Canopy Height Model | Maps the height of the canopy layer as a continuous function [61] |

### 2.3. Gradient Boosting Classification

Tree boosting models are widely used in the realm of machine learning due to their high effectiveness with state-of-the-art results in many real-world applications. In our preliminary studies, Gradient Boosting models consistently outperformed the popular machine learning algorithms. As shown in Figure 2, the data was applied to a range of other models, such as K-Nearest Neighbors (KNN), Decision Trees, Bagging, and Random Forest (RF). In our preliminary data analysis, KNN with K = 10 achieved an accuracy of 58.88%, Decision Trees an accuracy of 77.12%, Bagging an accuracy of 81.27%, and RF an accuracy of 81.76%, while Gradient Boosting methods achieve an accuracy of 83.20%.

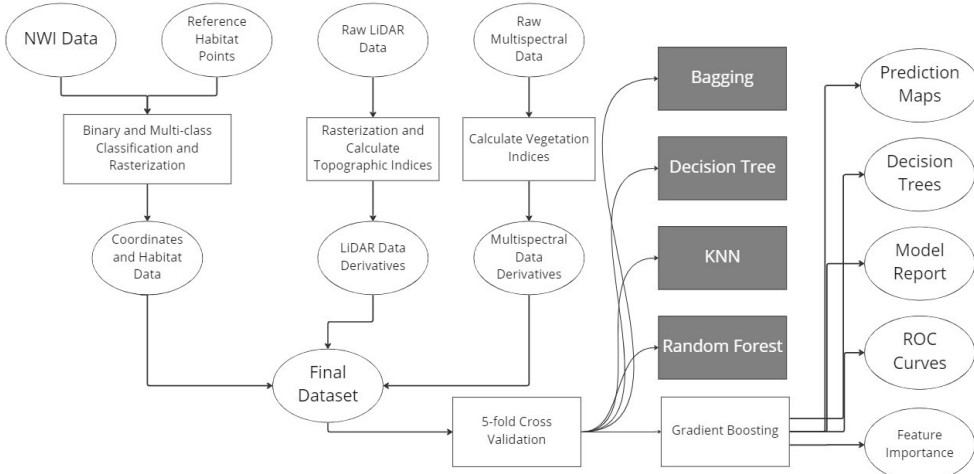

**Figure 2.** Workflow showing data preprocessing steps as well as the inputs and outputs of the Gradient Boosting models in addition to some other models tried (shown in grey and not used in analysis).

Therefore, three types of Gradient Boosting models were considered in this study: Gradient Boosting (GBM) [62], Xtreme Gradient Boosting (XGBM) [63], and Light Gradient Boosting (LGBM) [64]. These models create an ensemble of multiple decision trees while growing sequentially, fitting a modified version of the original dataset each time. Gradient Boosting models have the advantage of being able to handle substantial amounts of data and provide a high degree of accuracy.

XGBM improves the classical GBM in terms of high scalability for big data by several important systems and algorithmic optimizations, such as a novel tree learning algorithm to handle sparse data, the weighted quantile sketch for approximate tree learning, parallel and distributed computing with effective out-of-core tree learning. XGBM uses L1 and L2 regularization to smooth the final learned weights to avoid overfitting, and it implements parallel processing to greatly improve speed [64]. On the other hand, LGBM tackles two main challenges of efficiency and scalability in XGBM, especially in big data settings with high dimensional features and a large number of instances. LGBM aims to find solutions through the utilization of two novel techniques: Gradient-Based One-Side Sampling (GOSS) and Exclusive Feature Bundling (EFB). GOSS retains data instances with larger gradients while using random sampling on smaller gradients to contribute toward information gain. EFB reduces the dimensionality of the data to improve efficiency by bundling the features together [65].

These tree boosting models were applied to predict the wetland presence by using the habitat points of various sites with all the predictors obtained from the LiDAR and Multi-spectral data (Table 2). Figure 2 shows the data-preprocessing methods that comprised of creating the final dataset as well as the outputs from the final Gradient Boosting models. Hyperparameters were tuned to improve the classification accuracy with the consideration of low runtimes. The set of hyperparameters were listed as below: *n* estimators = 500, learning rate = 0.1, max depth = 6, loss = log loss, subsample = 1.0, criterion = Friedman Mean Squared Error, min samples split = 2, min samples leaf = 1, min weight fraction leaf = 0,

minimum impurity decrease = 0.0, ccp alpha = 0. The random state was kept constant in all models to avoid unexpected experimental variability.

The n estimators parameter measured the number of boosting iterations to perform. Since Gradient Boosting overfits at a lower rate than other methods as a result of the stagewise method of combining weak learners [66], we adopted the n estimators to be 500. Additionally, the max depth was set at 6 since it works well in the context of boosting [66]. The learning rate was set to 0.1 to shrink the contribution of each tree to the overall model. Since the amount of data was small, the subsample parameter was set to 1.0 since Stochastic Gradient Boosting was not necessary. The criterion was set as the Friedman Mean Squared Error due to its ability to compute the impurity of the current node and reduce it [63]. Each of the models was run using 5-fold cross-validation to assess its generalization performance and estimate the prediction accuracy of the models on new data. The following metrics were collected in all of the experiments: Runtime, Classification Report, Confusion Matrix, Accuracy, Receiver Operating Characteristic Curve (ROC), Area Under the ROC Curve (AUC) Score, Feature Importance, and the decision tree for visualization.

The decision trees are run for each datapoint in the input, which allows it to create a comprehensive output for a particular site. For each datapoint, the output is determined by following the path in the tree until a terminal node is reached. The value attached to the terminal node is then used to determine the classification of that specific pixel. Since gradient boosting is a generalized additive model (GAM), the results of each tree from previous iterations are combined to create an estimate of a "score" for classification. The score is then transformed using a logistic transformation, leading to the final classification.

## 3. Results

### 3.1. Individual Sites

GBM, XGBM, and LGBM were trained on each site separately using 5-fold cross-validation, and the trained model was used to test the pixel-based unlabeled data points for each site. Figure 3 shows the performance metrics of the GBM, LGBM, and XGBM on different sites: accuracy, sensitivity, and specificity. Sensitivity, which is also called the true positive rate, represents the probability of predicting as wetland given the true state is wetland. On the other hand, Specificity, which is also called true negative rate, represents the probability of predicting as non-wetland given the true state is non-wetland. Accuracy is the proportion of correct prediction among the total observations.

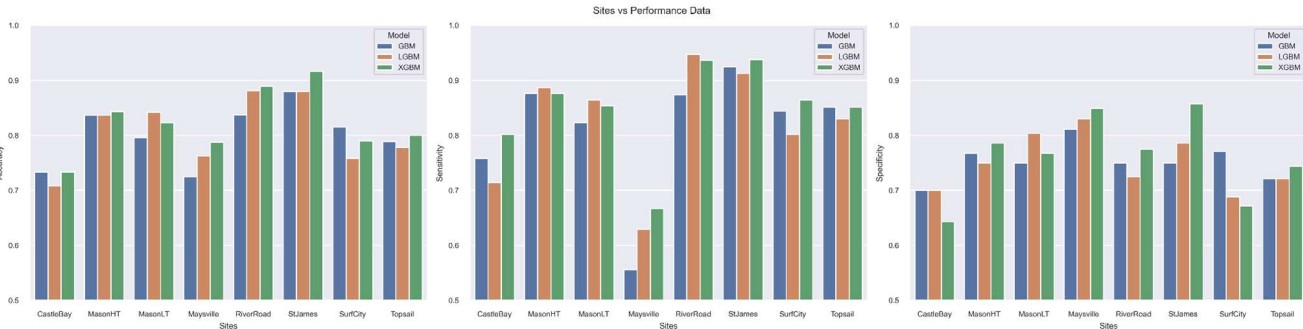

**Figure 3.** Performance metrics (Accuracy, Sensitivity, and Specificity in the order from left to right) of GBM, LGBM, and XGBM on each Individual Site.

As shown in Figure 3, the St. James site has the highest Accuracy (91.67%) and Specificity (85.71%) with the XGBM model, while the River Road site has the highest sensitivity (94.74%) with the LGBM model. On the other hand, for all three Gradient Boosting models, the Castle Bay site has the lowest accuracies (73.29% for XGBM and GBM; 70.81% for LGBM), while the Maysville site consistently has the lowest sensitivities (55.56% for GBM; 66.67% for XGBM; 62.96% for LGBM). Furthermore, the Castle Bay site and the Surf City site have overall low specificity for all three models. This was most likely a result of the size of the dataset, as Maysville had only 80 habitat points compared to the 108 points

of St. James and the 135–161 points of the other sites (excluding Topsail). Additionally, Maysville was the only site where the specificities were higher than the sensitivities for all three models. This was most likely because Maysville has a significantly higher proportion of non-wetlands than wetlands, while the rest of the sites have more wetlands than non-wetland. Lastly, Maysville and St James both have the highest specificities with the XGBM model. The reason why St James performed so well, despite not having the most habitat points, was likely because it had very few wetland types in the dataset (only PFO and PSS). This meant the model had less differences among wetland features to distinguish from in its final classification.

Supplementary Figure S7 shows the ROC curves for St James, Maysville, and Castle Bay as a model comparison, in addition to the AUC scores which serve as a summary to these ROC curves. The ROC and AUC are analyzed based on 5-fold cross-validation. For all ROC curves, a ±1 standard deviation was shown to account for the variability of individual folds.

For the AUC plot, the center represents the mean AUC, and the endpoints of the line segments represent the values of the AUC one standard deviation away. Across the various sites, GBM tends to have overall the highest variability, likely resulting from its greedy approach toward creating splits in the tree. Additionally, GBM tends to perform worse than the other models regarding AUC, similar to the other metrics of model accuracy. XGBM tended to perform the best across the sites, similar to its high performance across other accuracy metrics, cementing it as the model of choice.

St James' XGBM AUC score of 0.9545 makes it an ideal model, coming very close to an AUC of 1. Its second fold in Supplementary Figure S7a had a very low AUC, most likely a result of experimental variability, which caused the overall AUC to decrease. This was also the case for Maysville's XGBM ROC curve whose AUC was 0.8967.

The number of habitat points did not play a role in the ROCs or AUCs. As seen in Supplementary Figure S7c, Castle Bay had an AUC of 0.7752, the lowest among all the sites. While Castle Bay has the highest number of habitat points (161), its AUC is much lower than Maysville, which has only 80 habitat points. This is most likely because Castle Bay has six different types of habitats (the most out of any site), making it harder for the model to distinguish between wetlands and non-wetlands.

Another important metric that was considered across the sites was feature importance. Feature importance is a measure of how useful a variable is in the construction of the model. Variables used at the top of the decision tree contribute to the final prediction decision of a greater amount of input samples.

Supplementary Figure S6 and Table 3 show the intraclass homogeneity. Across all sites, a large mix of wetland types were present. The XGBM model trained individually on each site classified each wetland type as a wetland between 80% to 90% of the time, with the exception of PEM (which had a classification rate of 97%) due to it only being prevalent in one wetland site (River Road).

The St. James feature importance chart in Figure 4 shows that NDVI strongly correlates with the final model of the site and is more than three times more important than the next variable, CHM. This is easily visible in St. James' XGBM Decision Tree in Supplementary Figure S1a since the top-most node in the tree is a split based on NDVI which means it has the most influence on the output. The next most important splits are CHM and DSM, which are at depth two in the tree, also representing a strong influence on the output.

The Maysville feature importance chart in Figure 4 is also similar to St James due to the most important variable being twice as important as the next most important variable, sDTM. This is evident in Maysville's XGBM Decision Tree in Supplementary Figure S1b since DSM is the top node and thus has the most influence on the model's final output. This is followed by sDTM and Slope, which are towards the top of the decision tree.

**Table 3.** List of sites and most important variables when creating XGBM decision trees, ranked from highest to lowest.

| Site | Variables | Wetland Types |
|------|-----------|---------------|
| St James | NDVI, CHM, DSM | PFO, PSS |
| Topsail | NDVI, PlanCurv, CHM | PFO, PSS |
| Castle Bay | CHM, DSM, Slope | PFO, PSS, PEM, R5UB |
| River Road | sDTM, DTM, Slope | PFO, PEM, E1UB |
| Surf City | DTM, NDWI, NDRE | PFO, PSS, E2EM, E1UB |
| Masonboro High Tide | sDTM, hDTM, DTM | E1UB, E2EM |
| Masonboro Low Tide | sDTM, DTM, hDTM | E1UB, E2US, M2US, E2EM |
| Maysville | DTM, sDTM, Slope | PFO, R5UB |

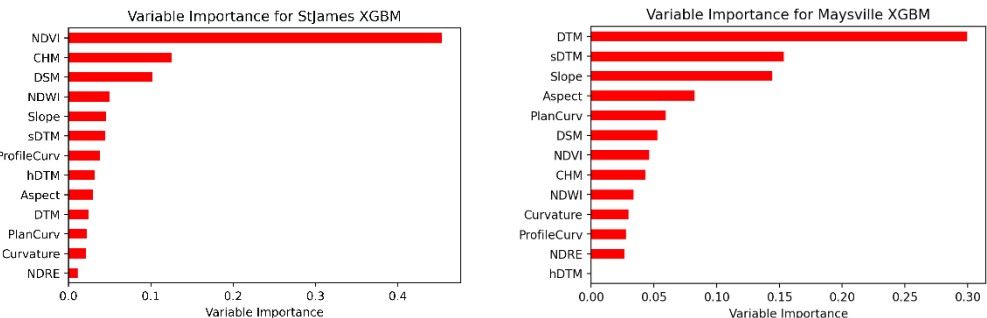

**Figure 4.** Feature importance for the XGBM Models of St. James and Maysville.

Using the Hyperspatial LiDAR data, a raster layer of the site's area of interest (AOI) is overlaid over a topographic map in geospatial software. This visualization uses RGB values to map the 3 most influential variables for each site, determined by the decision tree, with RGB color values. The blue pixels represent the CHM raster layer, the green layer represents DTM, and the red raster layer represents the NDVI values of the AOI. In Figure 5, the variable importance indicates that NDVI is the highest determinant for the St. James site. The graph on the right is the graphical representation of the XGBM classification map for the St. James site, where the green pixels represent wetlands, and the brown pixels represent non-wetlands.

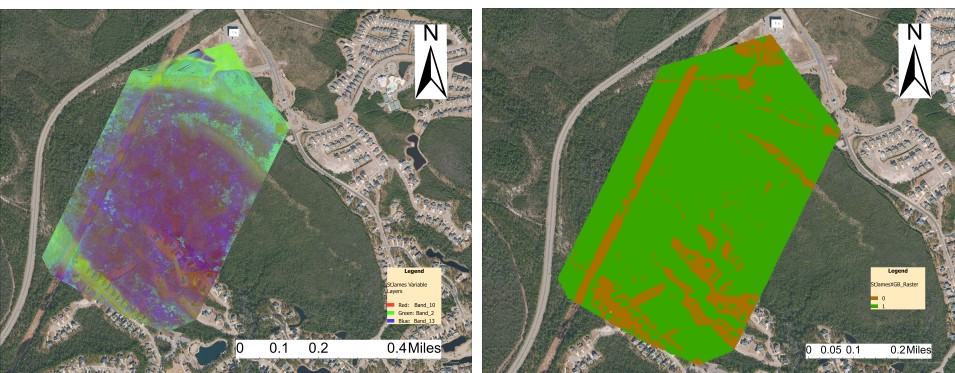

**Figure 5.** Visual representation of the spectral bands of the StJames site (**left**) and the XGBM classification model for the St. James site (**right**).

In the RGB raster map, pixels in the middle of the site were observed to contain high vegetation density (NDVI) and high values of CHM, which correlated to the classification of wetlands in the respective area. Although the green pixels show a high canopy layer around the edges of the St. James site, the absence of red pixels indicates smaller degrees

of NDVI and much sparser vegetation towards the outside of the study area. The DTM also contributes to the classification of wetlands, where areas with high DTM tend to be classified at non-wetlands, while areas with low DTM tend to be classified as wetlands.

Maysville's XGBM used Slope, DTM, and sDTM raster layers to classify areas, represented in Figure 6 (left). The highest performing variable was the DTM raster layer, which was represented by blue colors, followed by the sDTM raster layer represented by the green pixels, and the third highest performing variable was Slope, represented by the red pixels. The wetlands had a distinct correlation with areas of low elevations, shown by the distinct lack of blue and green pixels in the areas close to the center of the site in Figure 6. Areas around the wetland areas are elevated in height as shown by the prevalence of both green and blue pixels to the north of the site. These elevated regions also have red hues that signify steep slopes in the center of the site, signifying changes in elevation. Steep slopes leading to lower elevations leads to the pooling of water in water-saturated soils, and thus the formation of wetlands. The XGBM performed poorly on the Maysville site, over-classifying the non-wetlands and contributing to the XGBM's low accuracy with wetlands. Several factors may have influenced the model's performance including the low number of habitat points and the majority of the sampled points being classified as non-wetlands.

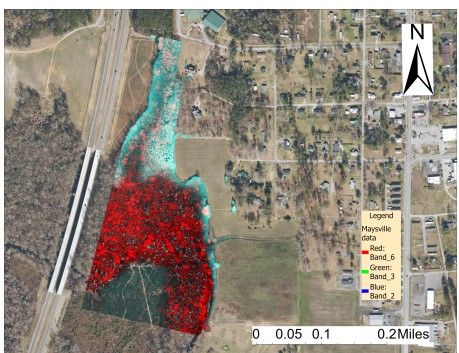 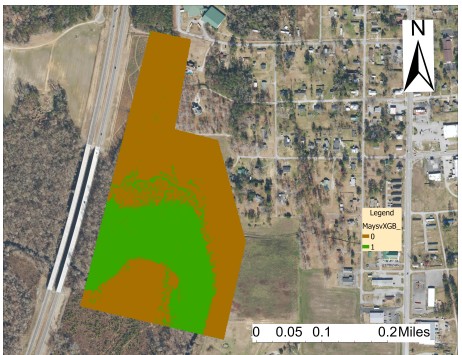

**Figure 6.** Visual representation of the spectral bands of the Maysville site (**left**) and the XGBM classification model for the Maysville site (**right**).

*3.2. Model Performance on Pooled Data with All Sites*

The data from all sites were pooled together to create a set of 1036 habitat points with all 12 covariates. All three models—GBM, LGBM, and XGBM—were built on the pooled dataset

Figure 7 shows the various accuracy metrics for the Gradient Boosting models. XGBM slightly outperforms GBM in accuracy, while both greatly outperform LGBM in accuracy. GBM displays the highest sensitivity, with XGBM as a close second, and LGBM performing the worst. XGBM has the highest specificity, greatly larger than both GBM and LGBM.

The ROC Curves in Figure 8 show that all three models had very similar shapes and variability. The ROC curves for individual folds also did not stray far outside of the ±1 standard deviation range. GBM had a slightly higher mean AUC than XGBM, while both models had a higher mean AUC than LGBM. GBM had the least variability in its AUC score, a stark difference from the large variability GBM had when trained on individual sites in Supplementary Figure S7. This means GBM likely can perform much better with larger sets of data, with a bigger increase in performance than XGBM and LGBM.

GBM and XGBM have very similar feature importance charts, while they both greatly differ from the LGBM feature importance chart (all shown in Figure 9). Both GBM and XGBM rank elevation-based variables like DSM, sDTM, hDTM, and DTM within the five most important variables for wetland prediction. They both also rank NDVI in the top five most important variables. On the other hand, Curvature-based variables such as PlanCurv, ProfileCurv, and Curvature are ranked among the less important variables in XGBM and GBM.

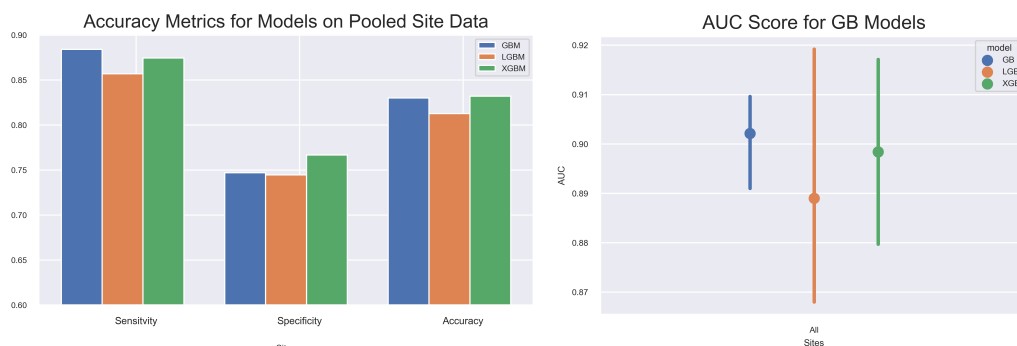

**Figure 7.** Performance metrics (Sensitivity, Specificity, Accuracy, and AUC scores) for three Gradient Boosting models on pooled data.

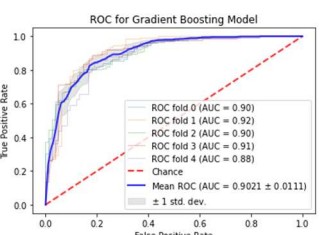
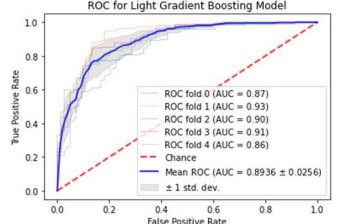
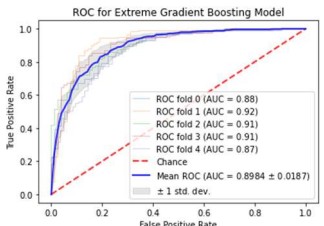

**Figure 8.** ROC Curves for all Gradient Boosting models on pooled data.

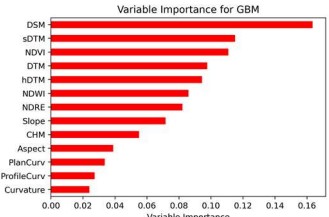
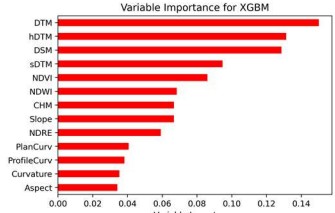
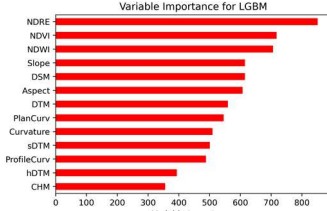

**Figure 9.** Feature Importance for all Gradient Boosting Models trained on pooled data.

LGBM was very different since the top three most important variables were those calculated based on the multispectral sensors: NDRE, NDVI, and NDWI, respectively. Slope is the fourth most important variable in the LGBM, although it ranks in the bottom half for both XGBM and GBM. LGBM, however, does have one elevation-based variable, DSM, in its top five. This discrepancy between LGBM and the other models could have been a result of LGBM's unique techniques: EFB and GOSS.

Supplementary Figure S2a shows a scatterplot of all the habitat points based on their DTM and NDVI separated by whether they are wetlands (colored in orange) or non-wetlands (colored in blue). Both wetlands and non-wetlands have similar distributions, although there are slight differences. For example, it seems there are two clusters in DTM for both wetland and non-wetland, while the pattern is more significant for wetland. Moreover, there are many wetland habitat points that have much lower NDVI values than non-wetland habitat points. Additionally, the wetland and non-wetland habitat points have different 'peaks' in their distribution for both DTM and NDVI, creating a difference that can be picked up by the gradient boosting models. Similarly, as shown in Supplementary Figure S2b, there are two clusters in sDTM for wetland. Furthermore, there is a linear trend in the sDTM vs. DSM plot. However, there are some differences in the distributions. For example, there is a difference in the value of peaks for both sDTM and DSM between wetlands and non-wetlands.

### 3.3. Model Performance on Pooled Data with Leave-One-Site-Out Scheme

In this section, we are interested in further examining the model performance on using the pooled data to predict for a new site, by proposing the Leave-One-Site-Out (LOSO) scheme. A single site is used for the validation set, and the remaining sites are pooled and made up the training set, except in the case of both Masonboro High and Low Tide datasets. To avoid information leaking, when testing on either Masonboro High Tide or Masonboro Low Tide, both datasets for Masonboro were removed from the training set to prevent overfitting. The same hyperparameters from previous sections were adopted to train these models.

As shown in Figure 10, one of the most surprising results was the vast discrepancies between high sensitivities and low specificities, or high specificities and low sensitivities. For example, Maysville had a sensitivity of 33.33% and a specificity of 81.13%. On the contrary, Topsail had a sensitivity of 93.62% and a specificity of 34.88%. The discrepancies could have been a result of the disproportionate distribution of wetlands and non-wetlands for these sites: Maysville had 53 non-wetland habitat points and only a meager 27 wetland habitat points, while Topsail had 27 wetland and 53 non-wetland habitat points. Additionally, both the Masonboro datasets had the highest accuracies, which came because of having both high sensitivity and specificity.

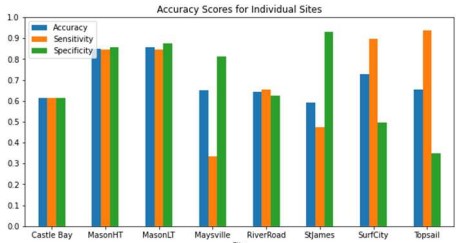 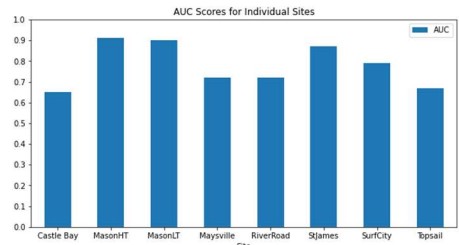

**Figure 10.** Performance metrics (Accuracy, Sensitivity, Specificity, and AUC scores) for XGBM.

There are some similarities in these results to Figure 3. For example, the Maysville site had a much lower sensitivity than specificity in Figure 3, and the same trend holds in Figure 10. Similarly, both Surf City and Topsail had higher sensitivities than specificities across both testing methods. However, there were some differences as well. St James had a much higher sensitivity than specificity when models were individually trained on a site but ended up having a much higher specificity than sensitivity when the models were trained on a pooled dataset. This could have been because there are two distinct wetland types within the St James Area of Interest (PFO and PSS) yet only one type of non-wetland, leading to a more accurate non-wetland classification.

As shown in Figure 10, the AUC scores, in general, were much lower than those in Supplementary Figure S7 when the model was trained and tested on a single site. This is similar to the trend for the other performance metrics since those also tend to be lower for cross site testing, when the model is trained on Leave-One-Site-Out pooled data compared to individual sites. An exception to this trend is the Masonboro Low Tide which had an AUC of 0.90 in Figure 10 but an AUC of 0.673 in Supplementary Figure S7. The respective ROC curves for XGBM on four sites, chosen in order from highest to lowest AUC, are shown in Supplementary Figure S3. The discrepancy for St James could have been caused by the unevenness of the distribution of habitat points, with 28 non-wetland points and 80 wetland points. On the other hand, Topsail and Castle Bay in Supplementary Figure S3c,d, respectively, had some of the lowest AUC values and thus the worst ROC curves out of all the sites. For some FPR values, the ROC curves did worse than random chance and led to the low AUC values. Topsail had low AUC values due to its large difference between sensitivity and specificity. Castle Bay had low sensitivity and specificity values overall, contributing to the low AUC.

Most sites tended to have similar top four important variables: sDTM, hDTM, DSM, and DTM. This was the most common in sites with PFO and PSS wetland types, as sites

with other wetland types had slightly different important variables. For example, the third most important variable in the Masonboro datasets was Slope, a result of the Estuarine wetlands it was composed of. Since estuarine wetlands tend to be formed around the base of slopes, it is expected that one of the most important variables is the Slope.

In Figure 11, both Castle Bay and St. James had the same list of the top six most important variables (in no particular order): DTM, DSM, sDTM, hDTM, NDVI, and Slope. Masonboro High Tide was slightly different, however, due to it including NDRE in the top six at the expense of hDTM which ranked the second-last. These feature importance charts were very similar to the XGBM Feature importance chart from Figure 9, but that was to be expected since both of them are based on pooled data.

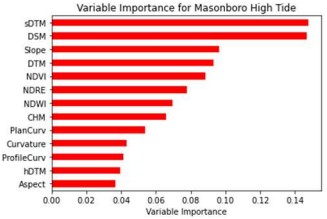 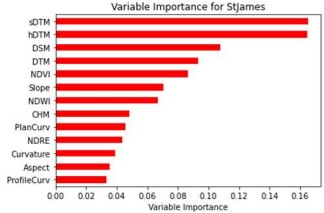 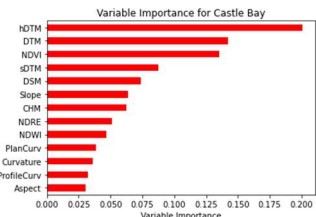

**Figure 11.** Feature Importance for Masonboro High Tide. St James, and Castle Bay for XGBM trained on Leave-One-Site-Out pooled data.

The most important variables in Table 4 are very different compared to Table 3 when the model was not trained on a pooled dataset. For example, out of the three most important variables of Masonboro High Tide in Table 3 (sDTM, hDTM, DTM), only sDTM was reflected in Table 4, with the others just being DSM and Slope. Similarly for Castle Bay, none of the three most important variables in Table 3 (CHM, DSM, Slope) were represented in Table 4 (hDTM, DTM, NDVI). Overall, the most important variables were similar to those presented in the XGBM Feature Importance but widely differed from the variable importance for individual sites in Table 3.

**Table 4.** List of sites and three most important variables when creating XGBM decision trees based on Leave-One-Site-Out pooled data, ranked from highest to lowest.

| Site | Variables | Wetland Types |
|---|---|---|
| St James | sDTM, hDTM, DSM | PFO, PSS |
| Topsail | hDTM, DSM, DTM | PFO, PSS |
| Castle Bay | hDTM, DTM, NDVI | PFO, PSS, PEM, R5UB |
| River Road | DTM, DSM, CHM | PFO, PEM, E1UB |
| Surf City | sDTM, DSM, DTM | PFO, PSS, E2EM, E1UB |
| Masonboro High Tide | sDTM, DSM, Slope | E1UB, E2US, M2US, E2EM |
| Masonboro Low Tide | sDTM, DSM, Slope | E1UB, E2US, M2US, E2EM |
| Maysville | hDTM, DSM, DTM | PFO, R5UB |

The decision trees in Supplementary Figure S4 largely align with the feature importance in Figure 11. For example, the root node of the Masonboro High Tide decision tree in Supplementary Figure S4a is based on the Slope variable, which has sDTM, DSM, and NDVI as children or descendants. Although DTM is the fourth most important variable for classifying Masonboro High Tide, it is not present in the decision tree. This could be because previous iterations of the tree could have contained representations of the DTM in their nodes, making the GAM contribute to the feature importance. Additionally, sDTM and hDTM encode similar correlations to DTM due to the nature of the variables, so the decision tree was able to have high accuracy because of the widespread use of sDTM and hDTM.

Similarly, for the St James decision tree in Supplementary Figure S4b, hDTM is the root note and has NDVI and DSM as children. DTM and sDTM, the other two out of the five most important variables, are present as lower branch nodes. However, hDTM encodes similar information as DTM and sDTM, meaning that the tree can still function to a certain degree of accuracy. Lastly, in the Castle Bay decision tree in Supplementary Figure S4c, hDTM is also the root node and only has NDVI as a child. DTM, its second most important variable, is only present towards the branch nodes of the tree. This could be why the Castle Bay model has such a low AUC.

Supplementary Figure S5 shows the distribution of some of the most important variables for Masonboro High Tide and Castle Bay. For Castle Bay, the third most important variable was chosen as a replacement for the second most important variable because the pair of hDTM and sDTM were too similar in distribution. From the DSM and sDTM distribution in the Masonboro High Tide dataset (in Supplementary Figure S5a,b), wetlands tend to have a much lower maximum and mean than non-wetlands. However, wetlands happen to have similar minimum values to non-wetlands, giving wetlands a lower range as well. This is also supported by the hDTM distribution in Castle Bay in Supplementary Figure S5c, although to a lower extremity. The NDVI in Supplementary Figure S5d for wetlands tends to have a similar mean to non-wetlands in the Castle Bay dataset, although there is a lower range and higher minimum. The indistinguishability of many of Castle Bay's most important features could have been the primary reason behind its low accuracy and AUC scores.

Compared to the St. James individual dataset in Figure 5, the pooled XGBM in Figure 12 performed worse in terms of sensitivity. The pooled XGBM for the St. James site did not reclassify any of the non-wetlands, leading to the overclassification of non-wetlands. The disparity between the two XGBMs can best be described by the difference in the decision trees. Whereas the individual site for St. James used NDVI, CHM, and DSM as their classification method, the pooled data relied more on DTM derivatives to classify wetlands. Since the different types of DTMs are all closely related as seen in the joint plot, this led to an inconsistency between individual and pooled XGBMs. Sites that performed well using DTM derivatives would continue to perform well, while sites that relied on other variables to improve the sensitivity of the model, would experience a decline in sensitivity. The St. James site's pooled XGBM reclassified the center portion of the AOI as a non-wetland, relying on the low elevation of sDTM, hDTM, and DSM.

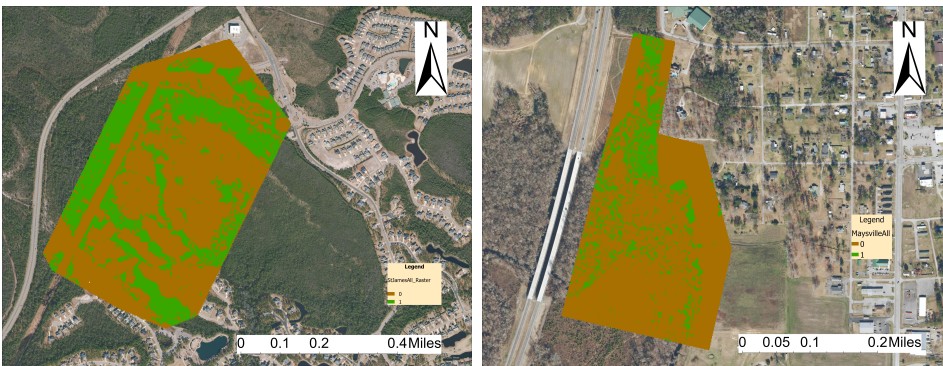

**Figure 12.** XGBM classification model for the StJames pooled Leave-One-Site-Out (**left**) and XGBM classification model for the Maysville pooled Leave-One-Site-Out (**right**).

Compared to Maysville's individual dataset in Figure 6, the pooled Leave-One-Site-Out XGBM in Figure 12 (right) was able to more accurately define the boundaries of wetlands using the variable raster data, but the XGBM model led to a more fragmented classification. The majority of the wetlands from the individual model were reclassified into non-wetlands, but non-wetlands further north were also reclassified into wetlands. The Maysville site experienced a tradeoff of specificity from the pooled Leave-one-site-out XGBM with sensitivity from the individual XGBM. Most of all the pooled XGBM for the

other sites resulted in a sensitivity that outperformed the specificity of both St. James' and Maysville's individual XGBM.

## 4. Discussion

This study showed that using gradient boosting models such as GBM, LGBM, and XGBM to predict coastal wetlands using LiDAR and Multispectral data is a feasible approach, especially given larger sets of data with more than 1000 data points. XGBM and GBM consistently outperformed LGBM, but XGBM would be the most feasible model on larger-sized datasets due to its advantage over GBM regarding low runtimes due to its scalability [64]. The overall accuracy, sensitivity, specificity, and ROC of the pooled XGBM model were 83.20%, 87.44%, 76.66%, and 89.84%, respectively. The runtime for the XGBM was also promising, 4.31 s compared to the 3.10 and 9.53 of LGBM and GBM, respectively. However, the accuracy metrics of individual sites greatly vary due to differences in wetland type and the distribution of habitat points between these types. The distribution of habitat points also affected the sensitivity and specificity of sites. Sites with more non-wetlands (such as Maysville) or a large difference in distribution (such as St James) tended to have a higher specificity than sensitivity values while the rest of the sites tended to have greater or equal sensitivity than specificity values.

This study also found that the DTM-based variables (DTM, hDTM, DSM, and sDTM) collected from the LiDAR data in addition to the NDVI data from the MS data had the greatest impact on the model predicting wetlands. The correlation between the lower elevations and wetlands can be attributed to the runoff of water. Lower elevations tend to collect runoff from higher elevations, which leads to the unique aquatic ecosystems that develop there. Interestingly, both Slope and Aspect, which are variables that direct water flow, were not major influencers in most decision trees. The Aspect variable, which records the downslope direction of the pixel, was the least influential variable when pooling the data together. This was probably because each pixel was classified independently, instead of influencing the classification of the adjacent pixels.

Previous literature has mainly used Sentinel-1 and Sentinel-2 data, using variables such as Red, Green, Blue, and Near Infrared bands, polarization backscatter data, and topographic data [32,47]. Two significant studies stand out, Wen et al. (2020) used ensemble classifier methods for the classification of wetlands, and Sun et al. (2020) used Random Forest models to analyze SAR remote-sensing images for wetland classification. The random forest in Sun et al. (2020) achieved 82.7% accuracy when analyzing coastal wetlands [47]. Wen et al. (2020) introduced ensemble methods, where their XGBM and GBM achieved an accuracy of 74.26% and 69.69%, respectively [32]. Additionally, a deep forest classifier model used Synthetic Aperture Radar data and had a 97.25% accuracy rate [40]. However, this model used 85,534 pixels as training data on a single site. Therefore, it is unclear whether the high accuracy rate was a result of the size of the data or methodology.

This study had some limitations, mainly the very low amount of habitat points collected for training the model. If the dataset was larger, the models would have been able to adapt to the widely different types of wetlands present across all the sites. Additionally, the smaller amount of data could have led to the overfitting of the model, especially considering the maximum depth of the tree was 6 nodes. St. James and Maysville sites both had considerably lower habitat points compared to the rest of the sites. The low number of habitat points led to the overclassification of non-wetlands for the St. James and Maysville sites in the pooled dataset.

This study can be expanded greatly to improve the accuracy of predicting wetlands using XGBM and these variables. Future work could include a study with more data and a concentration on a specific type of wetland using the same variables. This could also incorporate training and testing data on a site that was recently surveyed by the NWI to get a larger ground truth dataset. Another avenue for exploration is the usage of deep learning models on this dataset to identify wetlands.

## 5. Conclusions

Today, wetlands are at great risk of degradation and must be protected to conserve their ecosystems. This relies on classification from the National Wetland Inventory. Unfortunately, wetland classification is often an intrusive process that is frequently ineffective due to constantly changing ground conditions. Alternative methods like aerial or satellite-based remote sensing have been used, but these methods are either time-intensive and/or cost-intensive. Additionally, ground cover features may block aerial views of the area of interest, leading to incomplete classifications. Thus, a UAS-based method of remote sensing is proposed in the study.

This study shows that using Gradient Boosting ensemble methods as well as LiDAR and Multispectral sensors with UAS-based remote sensing methods are feasible to be used in identifying and classifying coastal wetlands. Furthermore, the study specifically highlights the potential of Xtreme Gradient Boosting over other methods due to its overall accuracy rate of 83.20%, sensitivity of 87.44%, and specificity of 76.66%. The study was conducted on multiple wetland types across 6 sites with only 1036 training data-points. Further research across more training data-points is recommended since that would improve the accuracy rate and improve the robustness of the model. Future works can also include Hyperspectral Sensors instead of hyperspatial data. The greater magnitude of spectral bands in hyperspectral data can introduce new variables and increase the complexity of current GBMs, especially with very few training data points.

Additionally, the study considers using training data pooled from multiple sites on a single site and finds mixed results. Some sites (St. James, Surf City, Topsail) have over 90% values for some of their accuracy metrics, while other sites (Maysville, Castle Bay) have under 60% values for some of their accuracy metrics. Thus, future research on more wetland types and wetland sites is recommended to improve the model.

The study considers 12 variables based on hyperspatial and multispectral data. The study shows that DTM-based variables (DTM, sDTM, and hDTM), DSM, and NDVI have the highest feature importance on pooled data, meaning that they are the most effective in differentiating wetlands from non-wetlands. However, on individual sites, the most important variables vary, especially across various wetland types. Furthermore, a study of intra-class separability is recommended in future work to analyze various wetland types more.

This study's methodology can greatly help the National Wetland Inventory in the United States enforce the Clean Water Act and keep its data updated due to the efficacy and speed of this method. Additionally, this can lead to advancements in UAS and sensor technology specifically suited to classifying wetlands.

**Supplementary Materials:** The following supporting information can be downloaded at: https://www.mdpi.com/article/10.3390/rs14236002/s1, Figure S1: XGBM Decision Tree for St James and Maysville after training on individual sites; Figure S2: Joint plot showing the distributions of DTM vs NDVI and DSM vs sDTM for every habitat point; Figure S3: ROC Curves for Masonboro High Tide, StJames, and Topsail after training on LOSO pooled data; Figure S4: XGBM Decision Tree for Masonboro High Tide, St. James, and Castle Bay after being trained on LOSO pooled data; Figure S5: A violin plot showing the distribution of DSM and sDTM in Masonboro High Tide as well as the distribution of hDTM and NDVI in Castle Bay; Figure S6: Accuracy of pooled XGBM model for different wetland types; Figure S7: ROC Curves for St James, Maysville, and Castle Bay after training on individual sites with XGBM, as well as AUC scores for each site and model type.

**Author Contributions:** Conceptualization, N.G.P., A.M. and C.C.; methodology, S.G., A.J.L., A.C.M., N.G.P., A.M. and C.C.; software, S.G., A.J.L. and C.C.; validation, C.C.; formal analysis, S.G. and A.J.L.; investigation, S.G., A.J.L., A.C.M., N.G.P., A.M. and C.C.; resources, N.G.P.; data curation, N.G.P. and A.M.; writing—original draft preparation, S.G., A.J.L., A.C.M.; writing—review and editing, N.G.P. and C.C.; visualization, S.G., A.J.L. and N.G.P.; supervision, N.G.P. and C.C.; project administration, N.G.P. and C.C.; funding acquisition, N.G.P. and C.C All authors have read and agreed to the published version of the manuscript.

**Funding:** This research was funded by the North Carolina Department of Transportation (RP2020-04) for all data collection and the National Science Foundation (DMS-1659288). The authors Shitij Govil, Aidan Joshua Lee, Aiden Connor MacQueen were supported by the State of North Carolina's Summer Ventures in Science and Mathematics recurring program funding to the University of North Carolina Wilmington for FY 21-22.

**Data Availability Statement:** The data used in this study can be shared upon contacting the corresponding author.

**Acknowledgments:** The authors would like to thank the anonymous reviewers for their careful reading and valuable comments that significantly improved the quality of the paper. The authors would also like to acknowledge Jessica Gray, Bailey Hall, Yishi Wang, Michael Suggs for the initial mentorship and guidance. The authors would also like to thank Garrison Payne for his initial contribution to the project. Finally, we would like to thank Joanne Halls, Britton Baxley, Kerry Mapes, Jesse Scopa, Carter Eckhardt, and James Giddens for their contributions to the data collection.

**Conflicts of Interest:** The authors declare no conflict of interest.

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
