# Peer review of "Using Hyperspatial LiDAR and Multispectral Imaging to Identify Coastal Wetlands Using Gradient Boosting Methods"

_remotesensing, doi:10.3390/rs14236002_

Round 1
Reviewer 1 Report
In this paper the authors presented a UAS-based gradient boosting approach to wetland classification in coastal regions using hyperspatial LiDAR and multispectral (MS) data, implemented on a series of wetland sites in the Atlantic Coastal Plain region of North Carolina, USA.
This paper needs major revisions before considering for publication, as follows:
-First, the authors must follow the MDPI format. The format is not good. Also, no line numbers, which is necessary during review process. Go to the authors instruction, and download Remote sensing format, then correct your paper format, and do not delete line numbers.
- Second, the abstract is very long. From the beginning to ( in this study), is very long, with only backgrounds. You should reduce the background.
- The paper just applied Gradient Boosting models, without comparisons to other models. So, read recent papers, and try to compare to other new models.
- As notices, no more details about the applied methods, no pseudocodes. These details are necessary to re-implement the method.
- Unnecessary description can be removed.
- The conclusion does not support the finding of the study. You have to improve it.
Author Response
Thank you so much for your feedback and comments, we really appreciate them. Please see the attachment for a more detailed response

Reviewer 2 Report
Overall this is a very interesting paper. It is well researched, written up and structured. The paper is also quite well contextualised re: the value and importance of the approach. A few minot edits to the text perhaps.... for consistency ... uncrewed instead of unmanned throughout. The approach is one that could play a useful role in subsequent classification of the vegetation in a wetland area. I feel the summary and conclusions are a little short and whilst they cover the outcome of the research could also have been perhaps tied into a wider discussion of the methodology and application of the study. One thing - as a Geographer - I would like to have seen - is a study area map, some ground photographs of the study area, and some illustrations relating to the sensors and drone platforms - always useful. Maybe also some illustration of the shortcomings of the satellite/aerial photography. what about hyperspectral sensors on UAV platforms? Considered? or not considered? for various reasons? Maybe some mention worthwhile. But interesting and study has potential.
Author Response
We thank you so much for your feedback and comments. For a detailed response to your comments, please see the attached file.

Reviewer 3 Report
Interesting work and research with new information for the research. Minor corrections in the work are suggested. In chapter 2.1 it would be good to add an image of the geographical area, chapter 2.2 Data collection and pre-processing are well described in the paper. However, chapter 2.3 should be improved and additionally provide a more illustrative description of the methodology. It is suggested that you may use workflow diagram techniques in this part of the paper. Also, part of the discussion (Table 3 and Figure 3) needs to be extended for comparative analysis for other sites and maybe use some of the appropriate diagrams or techniques for comparative analysis results and conclusions.
Author Response
We thank you so much for your feedback and comments. Please see the attached file for a more detailed response.

Reviewer 4 Report
In this manuscript, “Using Hyperspatial LiDAR and Multispectral Imaging to Identify Coastal Wetlands using Gradient Boosting Methods”, uses gradient boosting models to classify wetlands. The authors combined drone-based LiDAR and multispectral data to map wetland. Although the manuscript is in good structure, the manuscript overall does not yield much insights in this research field either in terms of methodology or application. I’m not aware of the knowledge gap that the authors were trying to fill. I’m afraid I have to reject this manuscript.
Some minor comments:
1. The key words: the manuscript uses “UAS” not “UAV”.
2. In “1.introduction”, the forth line has”,” after “.”.
3. Pay attention to the title of paragraphs, “2.1.” or “2.1”?
4. Table 2, the formulas about NDVI and NDWI are different in font size.
5. In “2.3”, pay attention to the indention of the last paragraph.
6. In”3.1.”, the sixth line has two “.”.
7. Figure 2, the size of four pictures is different.
8. The manuscript mentioned “SM Fig.1a” under Figure 3, where is the “SM Fig.1a”?
9. Figure 4, the legend in the right is same to Figure 5? And adding compass and proportional scale is easy to understand.
10. Figure 7 and 8 are lack of “.”.
11. Figure 9 has space character before “.”.
12. In “4.Disscussion”, the last line of the first paragraph is lack of ”.”.
Author Response
Thank you so much for your response and feedback. Please see the attachment for more details.

Reviewer 5 Report
Dear Authors,
The approach is mathematically correct but statistically not acceptable. The number of samples is to little to validate the results. The input data has to be better described in terms of acreage and spatial resolution of multispectral images. How they have been geommetrically corrected and radiometrically calibrated? How the indices have been calculated? From raw data or normalized radiances? Etc. Please see my comments in the file attached.

Author Response
Thank you so much for your response and feedback. Please see the attachment for a detailed response.

Round 2
Reviewer 1 Report
The authors addressed all comments.
This version can be accepted for publication.
Author Response
We thank you for your feedback throughout the process!
Reviewer 4 Report
The manuscript has been improved and it can be accepted now.
Author Response

(The authors gave the same response as above.)

Reviewer 5 Report
Please see my comments in the pdf attached.

Author Response
We thank you for your feedback. Please see the attached document for more details
